# Age-Dependent Pleomorphism in *Mycobacterium monacense* Cultures

**DOI:** 10.3390/microorganisms13030475

**Published:** 2025-02-20

**Authors:** Malavika Ramesh, Phani Rama Krishna Behra, B. M. Fredrik Pettersson, Santanu Dasgupta, Leif A. Kirsebom

**Affiliations:** Department of Cell and Molecular Biology, Biomedical Centre, Box 596, SE-751 24 Uppsala, Sweden; malavika.ramesh.2710@gmail.com (M.R.); prk.behra@icm.uu.se (P.R.K.B.); bmfpettersson@hotmail.com (B.M.F.P.); santanu.dasgupta@icm.uu.se (S.D.)

**Keywords:** pleomorphism, *Mycobacterium monacense*, non-tuberculous mycobacteria, DnaK3, MreB

## Abstract

Changes in cell shape have been shown to be an integral part of the mycobacterial life cycle; however, systematic investigations into its patterns of pleomorphic behaviour in connection with stages or conditions of growth are scarce. We have studied the complete growth cycle of *Mycobacterium monacense* cultures, a Non-Tuberculous Mycobacterium (NTM), in solid as well as in liquid media. We provide data showing changes in cell shape from rod to coccoid and occurrence of refractive cells ranging from Phase Grey to phase Bright (PGB) in appearance upon ageing. Changes in cell shape could be correlated to the bi-phasic nature of the growth curves for *M. monacense* (and the NTM *Mycobacterium boenickei*) as measured by the absorbance of liquid cultures while growth measured by colony-forming units (CFU) on solid media showed a uniform exponential growth. Based on the complete *M. monacense* genome we identified genes involved in cell morphology, and analyses of their mRNA levels revealed changes at different stages of growth. One gene, *dnaK*_3 (encoding a chaperone), showed significantly increased transcript levels in stationary phase cells relative to exponentially growing cells. Based on protein domain architecture, we identified that the DnaK_3 N-terminus domain is an MreB-like homolog. Endogenous overexpression of *M. monacense dnaK*_3 in *M. monacense* was unsuccessful (appears to be lethal) while exogenous overexpression in *Mycobacterium marinum* resulted in morphological changes with an impact on the frequency of appearance of PGB cells. However, the introduction of an anti-sense “gene” targeting the *M. marinum dnaK*_3 did not show significant effects. Using *dnaK*_3-*lacZ* reporter constructs we also provide data suggesting that the morphological differences could be due to differences in the regulation of *dnaK*_3 in the two species. Together these data suggest that, although its regulation may vary between mycobacterial species, the *dnaK*_3 might have a direct or indirect role in the processes influencing mycobacterial cell shape.

## 1. Introduction

Ever since the identification of *Mycobacterium tuberculosis* (*Mtb*) and *Mycobacterium leprae* as the etiological agents of tuberculosis and Hansen’s disease (leprosy), respectively, different species of mycobacteria have been the subject of extensive investigations into their growth, metabolism, morphology, pathogenicity and response to stress. The ability of rod-shaped mycobacteria to grow as coccoids, branched and spore-like cells has been noticed in early studies, and it appeared that mycobacterial cells undergo morphological changes upon ageing [1,2,3,4,5,6,7,8,9,10]. Recent studies using modern technologies such as atomic force microscopy confirm that the slow-growing mycobacteria (SGM) *Mtb* shows variation in cell shape at a late stage of its growth [11]. Moreover, the SGM *Mycobacterium marinum* (*Mmar*) and *Mycobacterium avium* subsp. *paratuberculosis* (MAP) were reported to form spores while spore-like structures have been detected in *Mycobacterium smegmatis* mc^2^155 (*Msmeg*), *Mtb* and *Mycobacterium bovis* BCG and *Mycobacterium phlei* cultures [7,12,13,14,15,16]. Our understanding of the underlying molecular mechanism dictating changes and variations in cell shape among mycobacteria is, however, limited. To identify the general genetic machinery that might be involved in regulating changes in cell shape in mycobacterial cultures, we examined different mycobacteria that appeared to demonstrate pleomorphic variations at different stages of their growth and/or under stress. Here, we focus on the non-tuberculous (NTM) *Mycobacterium monacense* (*Mmon*).

*M. monacense* was first isolated in 1998 in Germany from the bronchial lavage of an 80-year-old patient suffering from multifocal lung carcinoma and insulin-dependent diabetes mellitus. It forms smooth, yellow, scotochromogenic colonies within 5–7 days after inoculation when grown on solid media and *Mmon* belongs to the rapidly growing mycobacteria (RGM) [17]. *Mmon* has also been isolated from patients suffering from lung infections, nodular lesions or open wound infections [18,19,20,21,22]. During a microscopic survey of various mycobacterial species, we noted that the *Mmon* type strain DSM44395, referred to as *Mmon*^T^, changed its cell shape in an ageing culture. Hence, we decided to investigate growth, morphological characteristics and search for genes that might influence the cell shape of this RGM.

Here, we present data that *Mmon*^T^, and *Mycobacterium boenickei* (*Mboe*), cultures in liquid medium showing bi-phasic exponential growth curves obtained from absorbance. We determined a roughly four-fold difference in generation time comparing “early” and “late” exponential phases. However, for *Mmon* the generation times obtained by counting colony-forming units (CFU) did not differ at early or late phases of growth. Microscopy analysis of cells from different stages of growth showed changes in cell morphology from rod-shaped cells to coccoid and spore-like PGB cells upon ageing. This offered a probable explanation for the apparent bi-phasic nature of the growth curve plotted using absorbance data as cylindrical and smaller spherical shapes would scatter light differently [23]. To examine if there were any changes in the expression of genes associated with the growth-dependent changes in cell shapes, we identified genes predicted to be involved in the maintenance of cell shape and cell division using the complete *Mmon*^T^ genome. Their mRNA levels were monitored by RNASeq over time and the *dnaK*_3 mRNA level increased significantly upon ageing. DnaK_3 is a known heat shock chaperone and analysis of its protein architecture identified a domain of *Mmon*^T^ (and *Mmar*) DnaK_3 as an MreB-like homolog. Overexpression of *dnaK_3*^Mmon^ appeared to be lethal in *Mmon*^T^ but its exogenous expression in *Mmar* resulted in transient changes in cell morphology. Together, our findings provide a basis for discussing a possible role of DnaK_3, whether direct or indirect, in the regulation of cell shape in mycobacteria.

## 2. Materials and Methods

### 2.1. Bacterial Strains and Constructs

The wild type strain DSM44395 (*Mmon*^T^) was obtained from DSMZ (Deutsche Sammlung von Mikroorganismen und Zellkulturen, GmbH) culture centre, Braunschweig, Germany. The *Mmon*^RFPHyg^ and *Mmon*^RFPKan^ were constructed by transforming the *Mmon*^T^ with pDEAM5 and pDEAM2, respectively (permission number to work with *M. monacense* and introduce gene markers, 202100-2932v54a4, Uppsala University). These plasmids, which carry the red fluorescence protein (RFP) gene and the hygromycin or the kanamycin resistance genes, integrate at the *attB* site on the chromosome [24]. In order to over-express *dnaK*_3^Mmon^ in *Mmon*^T^, the gene was cloned into pBS401 plasmid (see e.g., [25]) behind a tetracycline-inducible promoter (pBS401-*dnaK*_3^Mmon^) and screened using plasmid specific-primers (Appendix A). Subsequently it was introduced into *Mmar* CCUG20998 (*Mmar*^T^) generating *Mmar*^pBS401-dnaK3Mmon^. A derivative of *Mmar*^T^ expressing the antisense *dnaK*_3^Mmon^ was also constructed, which is referred to as *Mmar*^pBS401-antidnaK3Mmon^. Details of the constructs, their nomenclature and primers used have been listed in Appendix A. For *Mmon*^T^ and *Mboe*^T^, all the experiments were conducted at 37 °C while in the case of *Mmar*^T^ the experiments were conducted at 30 °C.

### 2.2. Media, Determination of Generation Times and bi-Phasic Growth Experiments

MiddleBrook 7H9 (liquid) and 7H10 (solid) media were used as standard and prepared according to the manufacturer’s instructions. For cultivation of *Mmon*^RFPHyg^ and *Mmon*^RFPKan^, hygromycin and kanamycin were added to final concentrations of 100 µg mL^−1^ and 25 µg mL^−1^, respectively. For induction of *dnaK*_3^Mmon^ (and anti-*dnaK*3^Mmon^) carried by pBS401, 7H10 plates were supplemented with tetracycline (Tc; final concentration 20 ng mL^−1^) and hygromycin 100 µg mL^−1^ (see also Appendix A).

The Arret–Kirshbaum agar (AK; BD, Bioscience, Gothenburg, Sweden), modified G media (mG; [26]), Potato Dextrose Agar (PDA; BD, Bioscience, Sweden) [27] and Luria–Bertani (LA) media were prepared according to established protocols.

To determine the generation time, GT, for *Mmon*^RFPHyg^ cultivated in liquid medium a single colony (about 5–7 days old) was used to inoculate in 10 mL 7H9 medium supplemented with hygromycin (100 µg mL^−1^) and grown for 5–10 days. This primary culture was diluted 1000-fold in fresh media, incubated at 37 °C and OD_600_ was measured at different time points (see Section 3). Growth curves were generated by plotting OD_600_ values (*Y*-axis) against time (*X*-axis). The generation times were determined by calculating the slope for each curve using the tangent to the curve (exponential trend-line). The slopes (GT) were used to compare the growth phases (fast, slow or stationary) allowing us not to base our comparisons to the initial OD_600_ of the culture (see Section 3). Determination of the growth curve for each condition was repeated twice with biological triplicates for each repeat. The generation times were calculated based on two repeats with biological triplicates for each repeat and given as average ± error range.

The bi-phasic growth experiments were conducted in 7H9 (see above). A 5–10 days old primary culture of *Mmon*^RFPHyg^ grown at 37 °C was diluted 1000-fold (final volume 120 mL per replicate). OD_600_ was measured at regular intervals and plotted against time. From the so obtained bi-phasic growth curve, we identified a fast exponential phase Expo I, followed by a slow exponential phase Expo II and stationary phase SP (see Section 3). A fixed volume of Expo II cells (40 mL) was pelleted (in 50 mL falcon tubes) at OD_600_ ≈ 0.25–0.4 (for *Mboe*^T^ OD_600_ ≈ 1), and subjected to different media conditions, while the remaining culture was allowed to continue to grow (see Section 3). To test for the depletion of the carbon source, Tween 80 (final concentration 0.05%) and Glycerol (final concentration 0.2%) were added individually to 40 mL cultures in the Expo II phase. Media from the Expo II growth phase cultures were filtered using sterilised filters (0.2 μm; referred to as “spent media”) and used to re-suspend cells from Expo II (referred to as “Re-sus”) and Expo I (referred to as “EP-I in spent media”). As a control, cells from Expo I and Expo II were re-suspended in fresh 7H9 media (supplemented with 100 µg mL^−1^ hygromycin), referred to as “Expo I in fresh media” and “Expo II in fresh media”, respectively. Also, Expo II cells were pelleted and re-suspended in the same media (without filtration) as a control for the re-suspension step (and referred to as “re-suspended”). The OD_600_ was measured at regular intervals and growth curves were generated (for details see Section 3). At selected time points, cells were withdrawn and subjected to phase contrast microscopy and the percentage of cells with different cell morphologies were determined as described below. The bi-phasic growth experiments were performed at 37 °C.

For each condition, the experiment was repeated twice with biological triplicates for each repeat. The calculated generation times are given as average ± error range.

### 2.3. Determination of Growth Rates for Cells Grown on 7H10 Solid Media

Cells from a one-week-old plate grown at 37 °C were re-suspended in 2 mL 7H9 media to an OD_600_ of ≈0.5–0.7. A ten-fold dilution of this suspension was made and 100 μL of this suspension was plated on 7H10 plates (2–3 plates for each replicate) resulting in a final number of cells per plate corresponding to OD_600_ ≈ 0.07. Using the wide end of a Pasteur pipette, small areas of the agar was withdrawn from the plate after different times of growth and re-suspended in 1 mL of 7H9. This re-suspension was vortexed thoroughly, serially diluted and followed by plating of 100 µL on 7H10 of each dilution and incubated at 37 °C. After ≈5 days, single colonies were counted and the average colony-forming units (CFU) mL^−1^ for each time point was calculated and plotted against time. The generation time was determined from the slope of the exponential part of the so obtained plot.

### 2.4. Phase Contrast Microscopy

Small amounts of cells were scraped from plates and re-suspended in ≈500 µL of PBS. The cell suspension was diluted to obtain an almost clear suspension to have the appropiate amount of cells for microscopy. FM4-64, Mito Tracker Green (MTG) and DAPI (4′,6-diamidino-2-phenylindole) stains (Invitrogen, Carlsbad, CA, USA) were added to 100 µL of the above cell suspension to final concentrations of 5 µg mL^−1^, 5 µg mL^−1^ and 3 µg mL^−1^, respectively, and allowed to stand at room temperature for about 5–10 min. In total, 10 µL of this suspension was placed onto a slide with evenly solidified agarose gel (1% in PBS) and allowed to dry. A glass cover slip was placed over the dried spot and viewed at 100× magnification with a Zeiss Axioplan2 microscope with a CCD AxioCam camera (Carl Zeiss AB, Stockholm, Sweden) linked to the Axiovision 4.7 computerised image analysis software. See Appendix A for more information on DAPI staining patterns of *Mmon*.

### 2.5. Statistical Analysis

For every time point, a number of non-overlapping fields were viewed under the microscope (phase contrast) and each field was considered as one image for that strain. For each image, the numbers of different cell morphologies (rod, coccoid and PGB cells) were determined. For every time point and condition, a minimum of 350 cells were counted. The sum of all these morphology types was considered as the total number of cells present in the field. The percentages of each cell type and standard deviation were calculated. The average frequency of the occurrence for each cell type was first calculated between biological replicates and then between the repeats of the experiment. These values were then plotted in a bar plot with error bars indicating the standard deviations. The average and standard deviation was calculated for the frequencies obtained from each repeat (2–3 times) of the time course experiment.

### 2.6. Preparation of Cells for Transmission Electron Microscopy (TEM)

In total, 1 mL of cells were pelleted and fixed by re-suspending the cells in 5 mL 0.1 M sodium cacodylate buffer supplemented with 2.5% glutaraldehyde and rotated at low speed at room temperature for 30 min to keep the cells in suspension. The fixed cells were stored in the dark at 4 °C until initiating preparation for TEM. For a detail description, see Ghosh et al. [14]; see also [28].

### 2.7. Sample for Illumina and Pacbio Sequencing

For PacBio and Illumina sequencing of *Mmon*^T^, cells were grown on 7H10 (≈50 plates) for 2–3 days at 37 °C, collected by “scraping” and re-suspended in 40–50 mL PBS followed by centrifugation. For PacBio sequencing of *Mmon*^RFPHyg^, a 1000 mL culture was grown in 7H9 supplemented with 100 µg mL^−1^ hygromycin at 37 °C to OD_600_ ≈ 0.3 to 0.5 and harvested by centrifugation. Chromosomal DNA was extracted as previously described [29] and used for sequencing.

### 2.8. Genome Sequencing, Assembly and Annotation

The genome of *Mmon*^T^ was sequenced at the SNP&SEQ Technology Platform (HiSeq2000 Illumina platform) at Uppsala University, Uppsala. Illumina sequencing of *Mmon*^T^ generated reads with an average read length of 100 bp, average coverage of 400x, and assembled using the ‘Soapdenovo assembler-version 1.05’ [30] giving rise to contigs with minimum of 200 bps in length. The complete genomes of *Mmon*^T^ and *Mmon*^RFPHyg^ were sequenced at the NGI-Uppsala genome centre (PacBio technology), Uppsala, Sweden. The mean read length in *Mmon*^T^ PacBio sequence was 8625 bp with mean coverage of 58x and for *Mmon*^RFPHyg^ it was 9307 bp and 74x mean coverage. The long reads of the *Mmon*^T^ and *Mmon*^RFPHyg^ PacBio generated sequences were assembled using the SMART portal 2.1 analysis HGAP3 pipeline v3 [31] and polished with Quiver v1 (Pacific Biosciences, Menlo Park, CA, USA). The assembly of the Illumina-sequenced genome was performed as previously reported (see e.g., [32,33]).

Genome annotations for *Mmon*^T^ (Illumina and PacBio) and *Mmon*^RFPHyg^ (PacBio) were conducted using the PROKKA ver 1.10 annotation pipeline [34]. All the coding sequences (CDSs) were predicted using the ‘Prodigal-version 2.60’ [35]; ribosomal RNAs and tRNA using the ‘RNAmmer-version 1.2’ and ‘tRNAScanSE’ ver 1.23 applications [36,37]. The ‘non-coding RNA’ genes were identified using the ‘RFAM database-version 12.0’ [38] along with ‘Infernal-version 1.1.2’ aligner [39]. For generating the subsystem classification data, all the protein sequences predicted by the PROKKA pipeline were mapped to the RAST database (https://rast.nmpdr.org/, last accessed on 19 July 2019) and functionally classified [40] using the RAST webserver to obtain the subsystem classification data.

### 2.9. dnaK_3 over-Expression—Culture Conditions and Procedure

Single colonies of *Mmar*^pBS401-dnaK3Mmar/Mmon^ (about 5–7 days old) were inoculated into 7H9 medium supplemented with hygromycin (100 µg mL^−1^), in biological duplicates, and allowed to grow for 7–10 days. From this primary culture, 100 µL was plated onto fresh 7H10 hygromycin (100 µg mL^−1^) plates and incubated for about 2–3 days (exponential growth, adjusting to solid medium [41]). The cells were then scraped, re-suspended in PBS (to a final OD of about 1) and 100 µL of the suspension was plated onto fresh 7H10 hygromycin (100 µg mL^−1^) plates (for un-induced) and 7H10 with hygromycin and tetracycline, Tc (for induced condition; 100 µg mL^−1^ hygromycin and 20 ng mL^−1^ tetracycline). The plates were incubated at 37 °C and samples were harvested at various time points and prepared for microscopy and qRT-PCR.

### 2.10. Protein Homology Analysis

HMM (Hidden Markov Model) family of MreB_Mbl (PF06723) was retrieved from the Pfam database and an HMM search (V 3.1b2 with default settings and threshold T: 45) was conducted to identify the MreB_Mbl protein homologs in *Mmon*. The corresponding homologs were identified using the Simple Modular Architecture Research Tool (SMART; http://smart.embl-heidelberg.de/; last accessed on 26 March 2018 and 9 July 2019) [42] and presented using iTol (interactive Tree of Life V 5.4; see Section 3) [43]. Moreover, the online ‘NCBI BLASTp search’ tool (https://blast.ncbi.nlm.nih.gov/Blast.cgi, lasted accessed on 9 July 2019) was used to identify percentage identity and protein homology.

### 2.11. RNA Extraction, cDNA Conversion, RNASeq and qRT-PCR Analysis

Cells (two biological replicates) were harvested from plates at different time points and RNA was extracted, purified and converted to cDNA as previously reported [41]. The RNA sequencing (RNASeq) was performed at the SNP&SEQ Technology platform using the Illumina technology at Uppsala University, Uppsala, Sweden. The sequenced short reads of paired ends with a read length of 100 bp were mapped to the reference genome *Mmon*^T^ by creating an index and aligning using the ‘bowtie2-version 2.2.4’ tool [44] and Tophat pipeline-version 2.0.13 [45].

Based on the aligned BAM files, read counts were calculated using the HTSeq (version 0.9.1) [46]. The normalisation and differential expression analysis was performed using the Deseq2 [47] *p* + adj values, i.e., statistical significance is represented by stars based on the p + adj value range (* *p* < 0.05, ** *p* < 0.01 and *** *p* < 0.001). Finally, the differential mRNA levels were generated using the ggplot2 R-package ver 3.4.0 [48] and the gene synteny plots using the genoplotR R-package ver 0.8.9 [49].

Primers and probe design, and conditions for quantitative real-time PCR (qPCR) are listed in Appendix A. A 16S rRNA standard curve was used for relative quantification and as endogenous control [41,50]. Each sample (two biological replicates) was assayed in triplicates and the primer and probe details are listed in Appendix A. The *dnaK*_3 probe was designed to detect both *dnaK*_3^Mmar^ and *dnaK*_3^Mmon^. However, the primers targeting the *dnaK*_3^Mmon^ was specific to the gene on the plasmid (pBS401-*dnaK3*^Mmon^). To distinguish between the transcript levels of chromosomal *dnaK*_3^Mmar^ and *dnaK*_3^Mmar^ from the plasmid (pBS401-*dnaK3*^Mmar^), a control qPCR was performed (see Section 3) to determine the levels *dnaK*_3^Mmar^ mRNA derived from the chromosome.

### 2.12. Preparation of Whole Cell Lysates for β-Galactosidase Assay

Mycobacterial cultures carrying the pIGn empty plasmid and pIGn with *dnaK_3-lacZ* fusions were grown until exponential (≈0.5 OD600) and stationary phase (≈4.5 OD_600_) and cells were harvested by centrifugation. The pellet was then re-suspended in 1 mL 10 mM Tris-HCl (pH 8), re-centrifuged and the supernatant was discarded. The cells were once again, re-suspended in 200–400 µL of 10 mM Tris-HCl (pH 8), transferred to a 2 mL screw-capped tube containing 100 µL 0.1 mm zirconia beads and chilled on ice for 2–3 min. The cells were disrupted using FastPrep24 bead-beater for 45 s at speed 6.0 and chilled on ice for 2–3 min before centrifuging the lysates for 3 min at maximum speed to remove cell debris. The supernatant was transferred to clean 1.5 mL Eppendorf tubes and used for β-gal assay and Bradford assay for estimating total protein concentration.

### 2.13. Bradford and β-Galactosidase Assay

Reagents and protocol for Bradford assay for estimating the total protein was followed as recommended by BioRad (Bio Rad Laboratories AB, Solna, Sweden). The reading was measured in technical triplicates and biological duplicates for each sample using a 96-well plate reader (spectrophotometer) at 574 nm wavelength. Bovine serum albumin (BSA) provided in the BioRad kit was used as the standard.

For β-galactosidase assay, CPRG (Chlorophenol Red β-Galactopyranoside) prepared in Buffer X (KH_2_PO_4_ 50 mM at pH 7.8 and MgCl_2_ 1 mM) was used as substrate. In total, 15 µL of the sample lysate, 135 µL of Buffer X and 30 µL of CPRG (final concentration 1 mM) were added to each well, respectively. Readings were taken for 90 min with a time interval of 15 min between each reading (0 to 90 min) using a 96-well plate reader (spectrophotometer) at 574 nm wavelength.

## 3. Results

### 3.1. Growth of M. monacense in Liquid Media: Influence of hyg^R^ and the Red Fluorescence Protein (rfp) Gene

The type of strain *Mmon* DSM44395 (*Mmon*^T^) forms round, smooth and pale yellow colonies after 5–7 days of growth on MiddleBrook 7H10 (solid; Fisher Scientific, Gothenburg, Sweden) medium at 37 °C [17]. *Mmon*^T^ was also grown on other solid media, viz., Arret–Kirshbaum (AK), modified G (mG; a modified medium based on one known to promote sporulation in *Bacillus subtilis* [26]), LA and Potato Dextrose Agar (PDA; for details, see Section 2). Growth on LA, PDA and mG plates was slower and poorer, but similar pale yellow to yellow colonies did appear after 15 days at 37 °C (Table 1; Appendix A). Moreover, *Mmon*^T^ appeared to grow poorly (if at all) in liquid media (7H9, LB or mG). Some growth was, however, apparent in liquid LB and mG cultures after five days of incubation but with heavy clumping that made estimation of growth rates by measuring OD_600_ unreliable (Table 2).

To track *Mmon*^T^ (and ensure absence of contamination), we introduced the hygromycin and kanamycin resistance genes linked to the red fluorescent protein (RFP) gene into the *attB* site on the *Mmon*^T^ chromosome (see below and Section 2). These derivatives are referred to as *Mmon*^RFPHyg^ and *Mmon*^RFPKan^, respectively (Appendix A). The *Mmon*^RFPHyg^ grew in 7H9 liquid media supplemented with hygromycin while the *Mmon*^RFPKan^ did not show any detectable growth in 7H9 containing kanamycin (Table 2). Since *Mmon*^T^ transformed with the non-integrative plasmid pBS401 carrying *hyg*^R^ (referred to as *Mmon*^pBS401^; Appendix A) grew in 7H9 media supplemented with hygromycin (Table 2) it seemed that the growth was promoted by the presence of hygromycin in 7H9 liquid media and not a consequence of the chromosomal integration of pDEAM5 at the *attB* site (see below). Also, growth of *Mmon*^RFPHyg^ in liquid 7H9 medium without hygromycin resulted in heavy clumping. This would be consistent with earlier observations where hygromycin resistance appeared to confer a growth advantage (compared to kanamycin resistance) upon mycobacteria in the presence of respective antibiotics [51]. Unless stated otherwise, the experiments below were conducted in the standard media, 7H9 or 7H10.

### 3.2. Bi-Phasic Pattern of Growth Curves Plotted from Absorbance Measurements

Since *Mmon*^T^ and *Mmon*^RFPHyg^ grew differently in liquid medium, we compared their growth rates on solid medium, by following their growth on 7H10 plates without and with hygromycin (100 μg/mL). Identical aliquots of bacterial lawns from plates smeared with equal number of cells and maintained at 37 °C were collected at intervals and re-suspended into 0.1 mL PBS buffer. The number of bacteria counted by plating were plotted as CFU/mL against time (see Section 2). Figure 1A shows the growth curves for *Mmon*^T^ and *Mmon*^RFPHyg^ grown on 7H10 agar plates over a period for >100 h. Trend lines (marked in red) were drawn to estimate generation times (GT). The growth curves are almost parallel showing very similar growth rates for the two strains; viz., with GT, 5.9 ± 0.5 and 5.1 ± 0.9 h for *Mmon*^T^ and *Mmon*^RFPHyg^, respectively. Moreover, the polymorphic cell patterns (see below) for the two strains with ageing of the plate cultures were similar. Hence, the observed *Mmon*^RFPHyg^ growth and pleomorphism was taken to represent that of *Mmon* in general and we used *Mmon*^RFPHyg^ to investigate growth in liquid 7H9 medium containing 100 μg/mL hygromycin (see Section 2).

In order to follow the growth curve of *Mmon*^RFPHyg^ in liquid cultures, single colonies were inoculated into 7H9 medium supplemented with hygromycin and the growth rate was measured following OD_600_ with time for over 150 h (Figure 1B; see Section 2). Attempts to estimate GT by drawing trend lines through the time points showed two distinct rates of exponential growth with GT = 5.3 ± 0.44 h during the early phase (Expo I) followed by a much slower exponential growth rate, GT = 22 ± 3.1 h (Expo II) as shown in Figure 1B (see Section 2). The early growth rate is similar to GT (5.1 ± 0.9 h and 5.9 ± 0.5 h for *Mmon*^RFPHyg^ and *Mmon*^T^, respectively) for growth measured by cell counts on solid media (7H10; see Figure 1A). Upon longer incubation, the cultures appeared to reach stationary phase at OD_600_ ~3 (Figure 1B and Appendix A). The bi-phasic growth could be attributed to the following: (i) depletion of a preferred nutrient such as carbon source resulting in a diauxic growth curve; (ii) accumulation of inhibitor(s) in the medium (after a critical concentration is reached) that reduced the growth rate; (iii) cell density-dependent quorum sensing; or (iv) inherent growth phase dependent heterogeneity in cell size/shape [52,53,54,55,56,57,58]. To distinguish between these possibilities, we tested several growth conditions for *Mmon*^RFPHyg^ including fresh inoculation into “spent” media from Expo II phase cultures, cultivation of “Expo II-cells” in fresh media, addition of fresh carbon sources (glycerol or Tween 80) and different dilutions of “Expo II-cells” (for details, see Section 2).

Addition of glycerol or Tween 80 to Expo II cultures did not change the growth rate. Therefore, the transition from Expo I to Expo II cannot be attributed to depletion of these carbon sources [scenario (i); Figure 1C; Appendix A; however, we cannot rule out that other carbon sources in the 7H9 media such as dextrose, which is present in the growth supplement Oleic Albumin Dextrose Catalase (OADC), might have an impact, but see below. Cells in the Expo I-phase continued to grow with “Expo I-growth rates” irrespective of the media added (filtered media from the Expo II-cultures, “spent media”, or addition of fresh media; Figure 1D and Appendix A; see Section 2). Moreover, filtered and washed “Expo II-cells” continued to grow with “Expo II-rates” when inoculated into “fresh media” or re-suspended in the same filtered “Expo II-media” (with no apparent change in cell density; Figure 1D). These findings suggested that alternative (ii) is unlikely to be the reason for the slower “Expo-II growth rates”. This also suggests that depletion of carbon source may not be a factor causing the bi-phasic nature as re-suspension of Expo II cells in fresh media (complete media with all carbon sources) did not result in the Expo I growth rates. When “Expo II-cells” were re-suspended into “fresh” or “spent media” (see above) at the same cell density, the growth rates were similar to the “Expo II-rates”. However, when “Expo II-cells” were diluted 1000-fold, with “fresh” or “spent media”, the growth rate returned to “Expo I-growth rates” (similar results were obtained using cells from the Expo I-phase; Appendix A). Thus, any role of the altered composition of the culture medium with growth, either depletion of nutrients or a rise in the level of toxins released by the cells, could be ruled out as the reasons for the bi-phasic nature of the growth curve. Rather, the increase in cell density as the culture grew appeared to be the regulating factor altering the cell shape and size, suggesting the possibility of high cell density-dependent control of cell size heterogeneity in the cell population as the probable reason for the observed bi-phasic growth pattern. The involvement of a cell-density-associated quorum-sensing signal in causing the transition from rod-shaped to coccoid cells seems to be a possibility.

To further investigate the bi-phasic exponential growth shown by *Mmon*^RFPHyg^ we analysed the growth pattern of the RGM *M. boenickei* (type strain DSM44677; referred to as *Mboe*^T^) for which our data show similar results as *Mmon*^RFPHyg^ (Appendix A) along with variations in cell shape as observed in *Mmon^T^* and *Mmon*^RFPHyg^ cultures (see below). The GT for *Mboe*^T^ growing in the Expo I phase was estimated to be 3 ± 0.1 h, while for cells in Expo II it was 11.7 ± 0.9 h. The addition of a carbon source (Tween 80 or glycerol) did not change the GT, nor did re-suspension (Appendix A), which is similar to the findings for *Mmon*^RFPHyg^ grown in liquid media. Hence, the apparent bi-phasic pattern of the growth curve estimated from absorbance is not specific to *Mmon*^RFPHyg^; rather, it might be a common growth pattern among mycobacteria. Thus, we entertain the idea that inherent cell size heterogeneity in growing cell populations might be a common characteristic and plausible reason for bi-phasic growth pattern among mycobacteria.

### 3.3. Variation in Cell Morphology in Ageing Cultures

To further understand the nature of the bi-phasic growth pattern we followed the cell shape distribution of *Mmon*^T^ and *Mmon*^RFPHyg^ after growth on 7H10 (solid) media at 37 °C for 2, 8, 14 and 30 days by microscopy. *Mmon*^T^ cells were stained for membrane (FM4-64) and DNA [DAPI (4′,6-diamidino-2-phenylindole)], while *Mmon*^RFPHyg^ was stained with mitochondria tracking green (MTG), which stains outer as well as internal membranes [59], and DAPI (see Section 2). For each sample, we sequenced 16S rDNA as an additional control to rule out contaminations.

Three different cell morphologies were observed when viewed under the microscope (Appendix A): rods, coccoids and cells that appeared Phase Grey or phase Bright (PGB). The average cell sizes, measured for a minimum of 100 cells for each morphology type, were as follows: rods, 1.8 ± 0.31 µm (including both short rods of average length 1.6 ± 0.26 µm and longer rods of average length 1.9 ± 0.3 µm) and coccoids with 0.8 ± 0.19 µm diameters. This provided a qualitative basis for distinguishing between the different morphologies. For simplicity, rod-shaped cells detected in old cultures, although they were shorter than the exponentially growing rods, were still considered under the “rods” category.

At early stages of growth (1–2 days), the larger fractions of *Mmon*^T^ and *Mmon*^RFPHyg^ cells were rod-shaped, closely followed by coccoid-shaped cells (Figure 2A–D). After three days, we detected increasing appearance of coccoid shaped cells while the cells classified as PGB started to appear at later time points (after eight days). The different time points were chosen based on their relevance to the growth phase as shown in Appendix A. Thus, time points from 1 to 3 days belonged to the exponential growth phase and the rest of the time points were different stages in the early (5–8 days) and late stationary phase (14 days and beyond).

A heterogenous cell population of rods and coccoids was also observed when *Mmon*^T^ cells were examined by transmission electron microscopy, TEM. For *Mmon*^T^, FM4-64 staining and TEM further suggested asymmetric septum formation (Figure 2E,F) in keeping with what has been reported for other mycobacteria [25,60,61,62,63,64]. Moreover, similar changes in cell shapes as a function of time were also detected when cultivated on solid mG medium (Appendix A). However, the frequency of distribution in mG medium pertaining to PGB structures differed from growth on 7H10 medium, *Mmon*^T^ (Appendix A) and *Mmon*^RFPHyg^ (Appendix A). We emphasise that a change from the rod shape to coccoid was also observed in *Mmon*^RFPHyg^ liquid cultures (7H9 medium; Appendix A).

As for *Mmon*, the *Mboe*^T^ cell shape changed from rod to coccoid upon prolonged incubation on 7H10 medium at 37 °C, and PGB cells were observed after 14 days of incubation (Appendix A). The apparent change in the average cell size was also detected in liquid *Mboe*^T^ cultures (7H9 medium; Appendix A). We conclude that these two mycobacteria change their cell shape upon ageing, as reported previously for other mycobacteria (see e.g., [33] and the refs. therein).

### 3.4. Spore-like PGB Structures in M. monacense Culture

Micrographs of *Mmon*^T^ and *Mmon*^RFPHyg^ showed the appearance of the PGB structures at later stages of growth (Figure 2A,B; yellow arrows). One month old cultures of *Mmon*^T^ grown on 7H10 solid media at 37 °C, which had the highest frequency of PGB (Figure 2C and Figure 3A), was examined by TEM. We observed “round-shaped” dark grey bodies that appeared to originate from “mother” cells, since some of the structures were partly surrounded by membrane-like structures resembling spore-like cells (Figure 3B; red arrows). Staining with FM4-64 visualised outer membranes while MTG also showed presence of internal structures (Figure 3C). Re-inoculation of cells from the month-old cultures on a fresh 7H10 medium resulted in growth and the re-appearance of rod-shaped cells with a decrease in the frequency of PGB structures (Figure 3D; see also Section 4). However, these PGB structures did not meet the requirements of endospores; they were neither heat resistant when subjected to wet-heat treatment at 65 °C, nor did they show presence of dipicolinic acid (DPA). At all different steps, we confirmed that we were studying *Mmon* by 16S rDNA gene sequencing. These data suggested that the *Mmon* spore-like PGB structures are different compared to *Mmar* and MAP spores [14,16].

### 3.5. Genome Analysis of M. monacense

Several *Mmon* draft genomes and the complete genome of the type strain DSM44395 (*Mmon*^T^; e.g., Bioproject id. PRJNA521103) are available (Figure 4A) [65,66]. As a complement to this, we sequenced *Mmon*^RFPHyg^ using PacBio technology with mean coverage of 74x. Whole-genome alignment of *Mmon*^T^ and *Mmon*^RFPHyg^ showed high similarity and confirmed the insertion of the *rfp* and *hyg*^R^ genes at the *attB* site. We used the two complete genomes (*Mmon*^T^ and *Mmon*^RFPHyg^) and draft *Mmon*^T^ genome to map and identify genes associated with cell shape maintenance, cell division or development of the cell membrane.

The predicted number of genes in the complete *Mmon*^T^ genome was 5864, including 5772 coding sequences (CDS), 6 rRNAs (two rRNA operons), 47 tRNAs, 1 tmRNA, 1 RNase P RNA and 37 non-coding RNAs (see also [66]). Of the 5772 CDS, 3557 were functionally classified in at least one functional category (Figure 4B; see also e.g., [29,67]). Among functionally classified genes, a large fraction (1809 genes, ≈51%) were categorised in the “Amino acid and Derivatives”, “Carbohydrates”, “Fatty Acids, Lipids, and Isoprenoids” and “Cofactors, Vitamins, Prosthetic Groups, Pigments” subsystems while 155 (≈4.4%) genes were classified in the subsystem “Virulence, Disease and Defense” (for comparison, of 3906 CDSs almost 9% are classified as virulence genes in *Mtb* [67]).

Together, these data provide new insights into the biology of *Mmon* and as such might be of assistance for the identification of biomarkers that could be useful in developing diagnostic tools and new drugs targeting mycobacteria. With respect to the genes involved in cell wall synthesis and cell division, see below; their location on the chromosome is depicted in Figure 4A.

### 3.6. Analysis of Selected M. monacense Gene Transcripts at Different Growth Stages

Of the 3557 functionally classified genes in *Mmon*^T^ ≈3.7% were categorised into the subsystems “Cell Wall and Capsule” (104 genes) and “Cell Division and Cell Cycle” (27 genes; Figure 4B). This is comparable to other mycobacteria, see, e.g., Refs [29,33,67,68]. To analyse the mRNA levels at different growth stages we focused on selected genes with relevance to cell shape and as such might be involved in various stages of the formation of the peptidoglycan, septum, divisome complex and arabinogalactan (Appendix A) [69,70,71,72,73,74,75]. The time points were chosen based on the frequency of occurrence of the different morphologies observed in order to have a more homogenous population (Appendix A). Albeit coccoids occurred at all growth phases, the majority of cells were rods at early time points (exponential, i.e., 2 to 3 days), coccoids between 5 and 14 days and PGB structures at ≈1.5 months (48 days; for viability details of old cells see Appendix A). Hence, we isolated total RNA from *Mmon*^RFPHyg^ cells at different growth stages, determined mRNA levels by RNASeq (see Section 2) and mapped the transcripts to the *Mmon*^T^ complete genome. On the basis of the log_2_-fold change (Appendix A), mRNA levels for the majority of genes involved in formation of cell wall components were higher in exponentially growing cells compared to mRNA levels detected in cells from old cultures (14 and 48 days; Figure 5A–C; Appendix A): cell envelope genes, *ftsW*_1, *pbpB*, *pbpI*, *ripA*_1, *ripA*_3, *fbpA*_1, *fbpA*_2, *fbpC*_3 and *dprE1*_2. The mRNA levels for some gene transcripts, *cwsA*, *fbpC*_4 and *fbpC*2_1, were higher in cells from older cultures while those for *ponA1*_1 and *dprE1*_1 increased initially with age followed by a reduced level in 48 days old cells (Figure 5A–C; Appendix A). For *ftsZ* and *wag31* (a *divIVA* homologue [71]) we detected lower mRNA levels in old cells compared to cells growing exponentially as would be expected (Appendix A). Noteworthy, we cannot distinguish whether the change in mRNA levels is due to transcription/expression or degradation (but see Section 4).

Mycobacteria are equipped with several serine threonine protein kinases, STPKs, and among these, *pknA* and *pknB* influence cell morphology [70,76]. For *Mmon*^T^, 18 were annotated as STPK genes and four among these lack the kinase domain (Appendix A); the mRNA levels for most of these were either lower in cells in older cultures or did not vary significantly between the growth phases of *Mmon*^RFPHyg^ (6 d, 14 d, 48 d; Appendix A and Appendix A). For *pknA*, the mRNA level is higher in exponential cells in keeping with what has been reported for *Mtb* [70], while for *pknB*_1 (positioned adjacent and downstream of *pknA*), we did not detect any significant change in the mRNA level. Moreover, the transcripts of *pknK* and one of the annotated *pknH* copies, *pknH*_5 (which lacks the kinase domain) were higher in the stationary phase than in the exponential phase (Appendix A; Appendix A). The function of *pknK* is not well-established although it has been shown to be higher during stationary phase and may be linked to secondary metabolite metabolism [77,78]. The *pknH* is known to be involved in the regulation of the *embCAB* operon, which encodes the membrane-associated ethambutol target arabinosyl transferase (EMB), and might control intracellular bacterial growth during infection [79,80,81,82,83,84]. Finally, we detected a significant increase in *dnaK*_3 mRNA levels, from 0.31-fold (log_2_) in exponential phase (6 vs. 3 days) to 2.5- and 4.4-fold (log_2_), in 14 and 48 days old *Mmon*^RFPHyg^ cells (Figure 5D; see below).

### 3.7. Identification of an MreB Homolog Within DnaK_3

A protein BLAST homology search (BLASTp) revealed that the *Streptomyces coelicolor* MreB (SCO2611) is ~30% identical with ~56% query coverage to that of *Mmon* DnaK3 (MMONDSM_00498; *dnaK*_3^Mmon^). On the basis of protein domain similarity to the MreB-like domain MreB_Mbl, this raises the possibility that *dnaK*_3 might function as an *mreB* homolog (Figure 5E; see Section 2). The actin homolog MreB is involved in attributing and controlling the rod shape of cells in both Gram-negative and Gram-positive bacteria, e.g., *Escherichia coli*, *Caulobacter crescentus* and *B. subtilis* [85,86,87,88,89]. Specifically, MreB_Mbl proteins are constituents of the bacterial cytoskeleton and have mainly been reported for rod-shaped cells, the MreB_Mbl proteins/*mreB* or *mreB*-like genes have not been identified in coccoid species or in mycobacteria [10,60,90]. In this context, it should be mentioned that the MreB inhibitor A22 [91] prevents growth of both *Mmon*^T^ and *Mboe*^T^, which makes it tempting to speculate that the MreB domain of DnaK_3 might influence cytoskeletal functions essential for viability of *Mmon*^T^ and *Mboe*^T^. However, the possibility that the MreB-like function is provided by some yet unknown protein cannot be ruled out.

DnaK is a well-known heat shock chaperonin with a Hsp70 protein domain [92,93], and its N-terminal domain shows MreB_Mbl domain architecture homology also in some other bacteria (Appendix A). Moreover, *E. coli dnaK* mutants were found to affect (directly or indirectly) cell shapes and division [94,95,96,97,98]. The mycobacterial *dnaK*_3 gene is localised together with the nucleotide exchange protein gene, *grpE*, the chaperone *dnaJ*_1, and the transcriptional repressor *hspR* (Figure 5F) [92,99]. The mRNA levels for these transcripts also showed increments parallel to that of *dnaK*_3 with ageing (Figure 5G; Appendix A) suggesting that *dnaK*_3, *grpE*, *dnaJ*_1 and *hspR* constitute a transcriptional unit in keeping with previous findings, see, e.g., ([100,101] and the Refs therein).

Three additional genes were annotated as *dnaK* in the *Mmon*^T^ genome, *dnaK*_1, *dna*K_2 and *dnaK*_4. We therefore included these in the transcriptome analysis. However, in contrast to DnaK_3, these proteins either lack the MreB-Mbl domain (DnaK_2) or did not have a complete MreB-Mbl domain (DnaK_1 and DnaK_4), which encompasses 355 amino acids in DnaK_3 (Figure 5E; based on the Pfam database the average length for MreB_Mbl domains is ≈313 amino acids and for DnaK_1 and DnaK_4 ≈ 200 amino acids are present that relate to the MreB_Mbl domain). Comparing the transcript levels for the *dnaK* homologs in *Mmon*^RFPHyg^ revealed that the mRNA level for *dnaK*_3 increased upon ageing, 2.5- and 4.4-fold dependent on ‘age’ (log_2_; see above), while *dnaK*_1, *dnaK*_2 and *dnaK*_4 mRNA levels were marginally higher in exponentially growing cells relative to older cells (Figure 5D; Appendix A).

### 3.8. Expression of dnaK_3^Mmon^ in M. marinum Resulted in a Transient Change in Cell Morphology

Morphological changes correlated with increased *dnaK*_3 mRNA levels upon ageing of the cell culture. Hence, to examine whether overexpression of *dnaK*_3 affect cell morphology we cloned *dnaK*_3^Mmon^ in plasmid pBS401 behind the inducible tetracycline (tet) promoter. The resulting plasmid (pBS401-*dnaK*_3^Mmon^) was introduced in *Mmon*^T^. However, we were unable to obtain any transformants after several trials, suggesting that an increase in copy number of *dnaK*_3^Mmon^ is probably lethal to *Mmon*^T^.

Since we could not overexpress *dnaK*_3^Mmon^ in *Mmon*^T^, we tried to introduce the pBS401-*dnaK*_3^Mmon^ construct into the rod-shaped *M. marinum* CCUG20998 type strain (*Mmar*^T^, see Section 2) to investigate whether a higher *dnaK*_3^Mmon^ expression affected the *Mmar*^T^ cell morphology. The *Mmon*^T^ DnaK_3 protein is 94% identical to *Mmar*^T^ DnaK_3 (gene identity CCUG20998_00587 [66]) and the gene synteny of the *dnaK*_3 transcriptional units in these two mycobacteria are identical pertaining to *grpE*, *dnaJ* and *hspR* (Figure 5F).

First, the natural frequencies of appearance of the various cell morphotypes in *Mmar*^T^ was compared with that of *Mmon*^T^ (see below) as control. For this purpose, a red fluorescence-tagged *Mmar*^T^ strain [41], similar to *Mmon*^RFPHyg^, was used and is referred to as *Mmar*^RFPHyg^ (see Appendix A). The transformation of *Mmar*^RFPHyg^ with pBS401-*dnaK*_3^Mmon^ was successful. The resulting strain, *Mmar*^pBS401-dnaK3Mmon^, was examined for patterns of morphotype variations with growth, which appeared to be opposite to the pattern for *Mmon*^RFPHyg^: *Mmar*^RFPHyg^ showed higher frequencies of PGB cells during the early time points (Figure 6A) contrary to the pattern observed in *Mmon*^RFPHyg^ (Figure 2). Consistently, the *dnaK*_3 mRNA transcript levels in the two species with respect to growth phase differed, with an increase in ageing *Mmon*^RFPHyg^ cells while a decrease was detected in *Mmar*^RFPHyg^ (Figure 5G).

Microscopy and statistics of the different cell morphotypes showed a decrease in the frequency of PGB structures after 24 h incubations (Figure 6B, un-induced) compared to the control (*Mmar*^RFPHyg^; Figure 6A bottom panel). Upon “tet-induction” however, there seemed to be a transient increase in the frequency of PGB cells (up ≈ 15%) compared to the un-induced condition (Figure 6B). Compared to *Mmar*^RFPHyg^ the overall frequency of PGB cells decreased in *Mmar*^pBS401-dnaK3Mmon^ irrespective of induced or un-induced conditions. Thus, the increased transcription of *dnaK*_3^Mmon^ in *Mmar*^RFPHyg^ resulted in a decrease in the frequency of PGB cells. As shown in Figure 6C, phase contrast microscopy (Figure 6C, left panel) and MTG staining of cells from these cultures (Figure 6C, middle panel) indicated formation of internal membrane structures. Examination of the *Mmar*^pBS401-dnaK3Mmon^ cells with TEM also showed and confirmed the appearance of internal membrane structures (Figure 6C, right panel). Introduction of an “antisense-*dnaK*_3^Mmon^” (*Mmar*^pBS401-antidnaK3Mmon^) targeting the chromosomal *dnaK*_3^Mmar^ gene in *Mmar*^T^ did not result in such changes in cell morphology (Appendix A) and appeared similar to *Mmar*^RFPHyg^ (Figure 6A). Moreover, a plasmid carrying *dnaK*_3^Mmar^ behind a ‘Tet’-inducible promoter (pBS401-*dnaK*3^Mmar^) in *Mmar*^T^ (*Mmar*^pBS401-dnaK3Mmar^; Figure 6D), showed similar patterns of morphological variations as seen for *Mmar*^pBS401-dnaK3Mmon^ under tet-induced/un-induced condition with a decrease in the frequency of PGB cells (cf. Figure 6B,D). Thus, a reduction in the frequency of PGB cells in response to over-expression of either *dnaK*_3^Mmar^ or *dnaK*_3^Mmon^ (induced condition) in *Mmar*^T^ was consistent.

The *dnaK*_3^Mmon^ and *dnaK*_3^Mmar^ transcript levels in *Mmar*^pBS401-dnaK3Mmon^ and *Mmar*^pBS401-dnaK3Mmar^, respectively, were verified by qRT-PCR (see Section 2). As a control, we determined the *dnaK*_3^Mmar^ (chromosomal) mRNA levels in the *Mmar*^pBS401^ strain (see Appendix A). These data showed that both *dnaK*_3^Mmon^ and *dnaK*_3^Mmar^ mRNA levels were higher at the early time point (Figure 6E, 1 d time point) both under un-induced and ‘Tet’-induced conditions followed by a decrease upon prolonged incubation. This revealed that plasmids carrying *dnaK*_3^Mmar^ or *dnaK*_3^Mmon^ were transcribed in *Mmar*. In addition, these changes in transcript levels correlated with the differences in the frequency of appearance of PGB cells discussed above (un-induced and induced conditions; Figure 6B,D). Continued incubation/induction beyond four to five days did not change the frequencies of different cell morphologies significantly, nor did we observe such transient changes in the controls (Appendix A).

Taken together, these data support the speculation that *dnaK*_3 might indeed have an influence on cell morphology, although it is unclear how, and may be considered to possess an MreB-like function in modulating cell shape and/or size (see also Section 4).

### 3.9. Difference in dnaK_3^Mmar^ and dnaK_3^Mmon^ Expression in M. marinum

In contrast to *Mmon*^RFPHyg^, mRNA levels of *dnaK*_3 in *Mmar*^RFPHyg^ were higher at exponential phase when compared to stationary phase (Figure 5G and Appendix A, in 48 days old *Mmar*^RFPHyg^ cultures *dnaK*_3 mRNA levels were higher than in exponentially growing cells) suggesting different regulatory pathways of the *dnaK*_3 expression in the two species.

To verify this, *dnaK*_3^Mmar^-*lacZ* and *dnaK*_3^Mmon^-*lacZ* fusions including ≈250 bps of the respective upstream regions were generated using the pIGn plasmid (see Section 2; Appendix A). These constructs were introduced into *Mmar*^RFPHyg^ and we determined β-galactosidase activities in exponentially growing and stationary cells. As shown in Figure 6F we did detect β-galactosidase activity for the *dnaK*_3^Mmar^-*lacZ* construct while for the *dnaK*_3^Mmon^-*lacZ* construct resulted in no activity relative to the controls. This indicated differences in the expression of DnaK_3^Mmar^ and DnaK_3^Mmon^ in *Mmar*^RFPHyg^ irrespective of growth phase. This difference in the basal levels of DnaK_3 expression could be part of the reason to the inverse correlation patterns between cell morphology and DnaK_3 expression in *Mmar* and *Mmon*.

### 3.10. Identification of Putative Promoters, Sigma Factors and Regulatory “Elements”

Comparison of the upstream regions (300 bps) of the *dnaK*_3^Mmon^, *dnaK*_3^Mmar^ and *dnaK*^MtbH37Rv^ revealed putative sigma factor binding sequences. SigH was reported to regulate the expression of *dnaK* in *Mtb* [102]. On the basis of this and sequence similarity comparing the upstream regions, we predict that SigH is also involved in regulating *dnaK*_3 expression in both *Mmon*^T^ and *Mmar*^T^ (referred to as SigH^P1^; Appendix A). In this context, in *E. coli* suppressors of a mutant lacking *dnaK* map in *rpoH*, the heat shock specific sigma factor [97].

Further analysis indicated that transcription of *dnaK*_3^Mmon^ might involve several sigma factors SigE, SigG, SigH and SigM and possibly also SigB (see Appendix A). The putative *dnaK*_3^Mmar^ promoter region included possible SigD, SigE, and SigH binding sequences where SigE and SigH being common to both of *dnaK*_3^Mmon^ and *dnaK*_3^Mmar^. We also predicted other regulatory regions upstream of *dnaK*_3 such as putative binding sites for the transcription repressor HspR, HAIR (HspR associated inverted repeat) [99] and two CRP binding sites (catabolite repressor protein; see also Appendix A) [103,104]. In this context, we also identified putative binding sites for RegX (see e.g., [105]) upstream of *dnaK*_3 in *Mmon*^T^ (Appendix A). Our RNASeq data for *Mmon*^RFPHyg^ showed an increase in *senX3* and *regX3* mRNA levels over time in *Mmon*^T^ in contrast to *Mmar*^T^ where the levels of these transcripts did not change with growth phase (Appendix A).

For *Mmon*^RFPHyg^, RNASeq data showed increased mRNA levels for some sigma factors, e.g., SigG, SigH2 and in particular SigL2 in late stationary cells (48 days old cells), while for SigB, SigE and SigM the mRNA levels did not vary much with time (Appendix A–G). Further comparison of sigma factor mRNA levels in *Mmon*^RFPHyg^ and *Mmar*^RFPHyg^ (Appendix A–J; note that Appendix A–J correspond to Appendix A–C in Ref [62] and were included for comparison) revealed that in exponentially growing *Mmar*^RFPHyg^ cells almost 40% of the total sigma factor mRNA transcripts is represented by SigA mRNA. Reaching stationary phase SigA mRNA level is reduced while the fraction of SigB and SigE transcripts increased and together represent ≈80% (Appendix A; see also [41,67]). In contrast, the fraction of SigB and SigE mRNAs for *Mmon*^RFPHyg^ is already high (≈45%) after three days of growth, while SigA mRNA is significantly lower (Appendix A–G; for details, see also Appendix A).

Taken together, these findings emphasise similarities but also differences that might rationalise the divergent expression of *dnaK*_3 in *Mmon*^T^ and *Mmar*^T^. Furthermore, comparing the RGM *Mmon* and SGM *Mmar* suggested that the sigma factor mRNA levels vary depending on mycobacterium indicating differences in the regulation of sigma factor genes in response to changes in growth conditions, which likely will have an impact on the expression of other genes including *dnaK*_3. In addition, the presence of multiple sigma factor binding sites in the upstream region of *dnaK*_3 might reflect control of its expression in response to diverse environmental stress signals.

## 4. Discussion

*Mmon* is described as an “acid fast bacillus” belonging to the rapid growing mycobacteria, RGM [17]. A handful of reports on clinical isolates of *Mmon* exist [18,19] but a detailed account of the link between cellular morphology and its life cycle is missing. Therefore, to obtain insights into the life cycle of an RGM we have characterised the *Mmon* complete growth cycle focusing on cell morphology, growth characteristics and gene expression patterns at different growth phases. While the size and pigmentation of colonies did not vary much in different solid media, the impact of media was more dramatic when we attempted to grow *Mmon*^T^ in liquid media (Table 1 and Appendix A). A well-dispersed, measurable growth in liquid media with reproducible doubling time of about 5 h could be obtained for *Mmon*^T^ only when the hygromycin B resistance gene was introduced into the cells (Figure 1 and Table 1; growth of *Mmon*^RFPHyg^ in liquid 7H9 medium without hygromycin B resulted in a clumped and undispersed growth. Although the ribosome is the major target for hygromycin B (or its metabolic degradation products) in the bacterial cell leading to inhibition of growth [106], it is unclear how this could enable growth of *Mmon*^RFPHyg^ in liquid media (but see [51]). Furthermore, monitoring growth in liquid media by absorbance showed two distinct rates of exponential growth, a ‘fast’ and a ‘slow’ phase with generation times of ≈5 h and ≈22 h, respectively. Similarly, bi-phasic growth in liquid culture was also detected for the RGM *Mboe*^T^ (Appendix A), suggesting that it is not unique for the RGM *Mmon*.

### 4.1. Variation in Cell Shape and Growth Phase

The possibility of reduced growth rate due to depletion of nutrients or accumulation of growth inhibitors in ageing cultures was discounted from resuspension experiments (see Section 3). But cell density in the culture appeared to be critical since growth rates could be adjusted by the cell density levels. Since growth on solid media, as measured by CFU, did not appear to suggest a bi-phasic growth pattern, changes in cell shape and/or size with age causing a shift in absorbance resulting in the observed bi-phasic growth in liquid media seemed plausible. Since coccoid shaped cells scatter light differently compared to rods [23] an increasing population of coccoid cells may manifest as apparent reduction in turbidity measurements and hence bi-phasic growth.

Genotypically clonal mycobacterial cells are known to exhibit phenotypic heterogeneity that is reflected in cell size, among other characteristics [25,61]. Transition from rod to coccoid or to smaller rods has been observed in mycobacteria isolated from infected cells or organs [107,108]. Nutrient starvation or microaerobic conditions have been reported to result in rod to coccoid transitions in laboratory cultures (liquid media/agar plates) [109]. Such a change in morphology has been associated with rise in the intracellular level of (p)ppGpp in *M. smegmatis* [110]. Also, reduction in size in stationary phase cells has often been attributed to division without growth undertaken by the bacterial cell as an adaptive measure against nutritional deprivation [111,112]. In the present study, however, *Mmon*^T^/*Mmon*^RFPHyg^ cells showed a heterogenous population of cells with different cell shapes. A shift in the frequency of occurrence from rods to coccoids as early as 3 days was observed, while the cells were still in their exponential growth phase (Figure 2 and Appendix A), suggesting factors other than stress or ageing as the cause for such natural pleomorphism. The cause for such a change from rod-shaped cells to coccoids (Figure 2) in *Mmon*^T^/*Mmon*^RFPHyg^ with increasing cell density is however, not clear. Moreover, the rod to coccoid transition in *Mmon*^RFPHyg^ with increasing cell density was reversed by dilution to the original cell density with either ‘fresh’ or ‘spent media’ (Appendix A) apparently without any help from factors released into media with ageing. How higher cell density generates this signal is not clear and whether such signalling is mediated through stringent control remains to be investigated.

The distribution of diverse cell types at different stages of growth from 2 to 30 days in *Mmon*^T^ and *Mmon*^RFPHyg^ are shown in Figure 2 [*Mboe*^T^ (Appendix A)]. Despite their appearance in phase microscopy, the PGB cells did not demonstrate heat resistance or DPA-content. However, use of membrane-specific stains (FM4-64 and MTG) and TEM revealed internal membrane structures and compartmentalisation (Figure 3). In addition, endospore staining (Schaeffer–Fulton stain) of cells from old cultures also indicated the possibility that these refractive structures could be endospore-like structures (Appendix A), supporting the possibility that mycobacteria such as *Mmon* might have evolved a pathway to undergo a transition from standard rods to more spherical/coccoid and/or spore-like PGB cell shapes. This pathway would be different from the well-known endosporulation process in Firmicutes, spore formation in Actinobacteria such as *Streptomyces* spp., and in the SGM *Mmar*^T^ and *M. avium* subsp. *paratuberculosis* in response to various stress/growth conditions [14,16,113,114,115,116].

### 4.2. M. monacense Genes Involved in Cell Division, Cell Shape Maintenance, Peptidoglycan Formation and Other Membrane/Cell Wall Related Functions and Their mRNA Levels

Based on the complete *Mmon*^T^ (and *Mmon*^RFPHyg^) genome we annotated genes involved in cell division, cell shape maintenance, peptidoglycan formation and other membrane/cell wall related functions (Appendix A). Our RNASeq data for *Mmon*^RFPHyg^ suggested lower mRNA levels in stationary phase than in exponentially growing cells for these genes with some PBP (Penicillin-Binding Protein) gene transcripts marginally higher. In contrast, the *dnaK*_3^Mmon^ (MMON_00498) mRNA level increased upon ageing reaching a 4.4-fold (log_2_) higher level at 48 days (Figure 5D,G; Appendix A). On the basis of protein domain similarity to Mbl (MreB-like protein; Figure 5E) we identified a domain in DnaK_3^Mmon^ as an MreB homolog and as in other bacteria *dnaK*_3 is positioned upstream of *grpE*, *dnaJ*_1 and *hspR* both in *Mmon* and in *Mmar*^T^ (Figure 5F) [117,118,119]. Also, *dnaK*_3 from these two mycobacteria is well-conserved (94% identity). The MreB and Mbl proteins are involved in determining the cell shape in different bacteria such as *E. coli* and *B. subtilis* [85,86,120,121,122]. These proteins have been reported to interact with PBPs, FtsZ and Rod in these bacteria [122,123]. Moreover, *E. coli dnaK* mutants show defects in cell division, chromosome segregation and filamentous growth ([94,95,96,97]; for the effects in D*dnaK* D*tig* double *B. subtilis* mutants see [98]). Studies of the chaperone DnaK in *B. subtilis* suggests that it is membrane associated and involved in cells’ recovery in response to certain stresses [124]. Moreover, higher levels of DnaK have been identified in germinating spores of *Streptomyces granaticolor* [125]. Data further suggest that DnaK has an essential, non-redundant role in cellular growth and protein folding in *M. smegmatis* [126]. Moreover, it appears that *Mtb*^DnaK^ binds to HspR when it is bound to HAIR (e.g., the HspR binding sites upstream of *Mtb dnaK*, which are also present upstream of *dnaK*_3 in *Mmon*^T^ and *Mmar*^T^; Appendix A) and as such influence the regulation of the *dnaK* operon [101,127,128].

### 4.3. Cell Shape and Possible Role of the DnaK_3 Chaperone

Apart from MreB, many bacteria also code for DnaK and DnaJ-like heat shock proteins with MreB_Mbl protein domains in addition to the Hsp70 domain (Appendix A). Mutations in such genes affect protein folding, temperature-sensitivity and growth rate to name a few [129,130,131]. Three-dimensional structure predictions and amino acid sequence motif comparisons of some functionally different proteins have shown that, despite their differences in function and low amino acid similarity, the folding of some proteins, e.g., DnaK/Hsp70, ParM, MreB and FtsA, are similar to that of actin proteins [132,133].

We could not successfully over-express the *dnaK*_3^Mmon^ in *Mmon*^T^, which might indicate that increasing the number of *dnaK*_3^Mmon^ gene copies is lethal to *Mmon*^T^. However, we provide data that show the introduction and expression of *dnaK*_3^Mmon^ in *Mmar*^T^ correlated with changes in cell morphology raising a possible role (direct or indirect) for *dnaK*_3 in cell shape maintenance. The observed changes in cell morphology were transient with respect to coccoids and PGB cells, which was also true for the expression of *dnaK*_3^Mmar^ in *Mmar*^T^. Similar pattern of occurrence for PGB cells was observed when *dnaK*_3^Mmar^ was over-expressed in *Mmar*^T^; however, there seemed to be no impact on the coccoid cell shape. These changes in cell morphology were not observed when a plasmid carrying an antisense-*dnaK*_3 (pBS401^antidnaK3Mmon^) gene was introduced into *Mmar*^T^ (Appendix A). In contrast to *Mmon*^RFPHyg^, the *dnaK*_3^Mmar^ mRNA (and *grpE*, *dnaJ*_1 and *hspR*) level is lower in cells in stationary phases compared to exponentially growing cells (Figure 5G; however, in late stationary cells, 48 days old cultures, we observed ≤1.3 log_2_-fold higher levels than in exponentially growing cells) and we observed many PGB cells during the early time points (1 day) after which the frequency decreases (Figure 6A). Together, the results indicate that *dnaK*_3 may be involved in cell shape maintenance indirectly or directly albeit, *dnaK*_3^Mmar^ and *dnaK*_3^Mmon^ appear to be regulated differently in these two mycobacteria (discussed below). We also emphasise that the MreB inhibitor A22 [91] prevented growth of both *Mmon*^T^ and *Mboe*^T^.

These differences in the frequencies of occurrence of the various morphologies observed in response to *dnaK*_3^Mmon^ vs. *dnaK*_3^Mmar^ (behind ‘Tet’ inducible promoters) over-expression in *Mmar*^T^ suggests possible differences in the regulation of *dnaK*_3 expression. This is supported by our data suggesting that the expression of the native *dnaK*_3 *Mmar*^T^ and *Mmon*^T^/*Mmon*^RFPHyg^ genes (with their respective upstream regions) fused to the *lacZ* gene differed when expressed in *Mmar*^T^ (Figure 6F). In keeping with this, our genomic analysis suggested that the *dnaK*_3 regulatory regions differ with respect to predicted promoter sequences in *Mmon*^T^/*Mmon*^RFPHyg^ and *Mmar*^T^; SigB, SigE, SigH, SigG and SigM promoters for *Mmon*^T^, and SigD, SigE and SigH promoters for *Mmar*^T^ (Appendix A). Also, we observed variations in sigma factor mRNA levels upon ageing comparing *Mmon*^RFPHyg^ and *Mmar*^RFPHyg^. It could be argued that the stability of the *dnaK*_3 mRNA in *Mmon*^T^ and *Mmar*^RFPHyg^ differ. However, our unpublished data suggest that the half-lives are similar, 5.6 ± 0.67 (*Mmon*^T^) and 5.0 ± 0.81 (*Mmar*^RFPHyg^) min (to be published elsewhere). Together, this indicates that the difference in the *dnaK*_3 regulatory region in these two mycobacteria might be the reason for the observed difference in the *dnaK*_3 mRNA levels in *Mmon*^RFPHyg^ and *Mmar*^RFPHyg^ as well as the appearance of a transient level of *dnaK*_3^Mmon^ mRNA when introduced into *Mmar*^T^. Nevertheless, with respect to documentations of mycobacterial cell morphologies at different growth stages and conditions with respect to *dnaK*_3 gene expression the data provide new insights and as such pave the way for deciphering the mycobacterial life cycle.

### 4.4. Concluding Remarks

The *dnaK*_3, *grpE*, *dnaJ*_1 and *hspR* constitute a transcriptional unit in the RGM *Mmon* and SGM *Mmar*, where *dnaK*_3 encodes for a chaperone belonging to a family of heat shock proteins. DnaK in, e.g., *E. coli*, has a key role in protein homeostasis. The *grpE* and *dnaJ*_1 encode for a nucleotide exchange factor and a co-chaperone, respectively, both having crucial functions in protein homeostasis while HspR represents a transcription repressor [93,134]. Both *Mmon* and *Mmar* carry several *dnaK* genes while *Mtb* has only one positioned in an operon together with *grpE*, *dnaJ*_1 and *hspR* [135]. It has been discussed that chaperones in *Mtb*, including DnaK, have moonlighting functions mainly related to virulence. For example, *Mtb* DnaK functions as a signalling molecule and in *Lactococcus lactis* DnaK binds the host cell invertase that subsequently influence its virulence [119,136,137]. Our findings that *dnaK*_3 might be a factor influencing cell morphology might be an additional example of a protein with moonlighting function in mycobacteria. However, at present we cannot exclude the possibility that the intracellular level of DnaK_3 affects the expression and/or function of other genes that are more directly responsible for the observed changes in cell morphology. In this context, *E. coli* DnaK is suggested to have a role in cell division and segregation and its deletion results in filamentous growth [95,97,138]. Moreover, DnaK is involved in the control and level of RpoS and RpoH (σ^32^) when *E. coli* is subjected to stresses such as heat shock and glucose starvation, and it binds to RpoH [139,140,141]. These scenarios might therefore also apply to mycobacteria.

In conclusion, our findings provide insights into the biology of *M. monacense* and the possible role of the chaperone DnaK_3. As such, they may be used in identifying biomarkers that could be useful in the process to develop diagnostic tools and new drugs targeting mycobacteria.

## Figures and Tables

**Figure 1 microorganisms-13-00475-f001:**
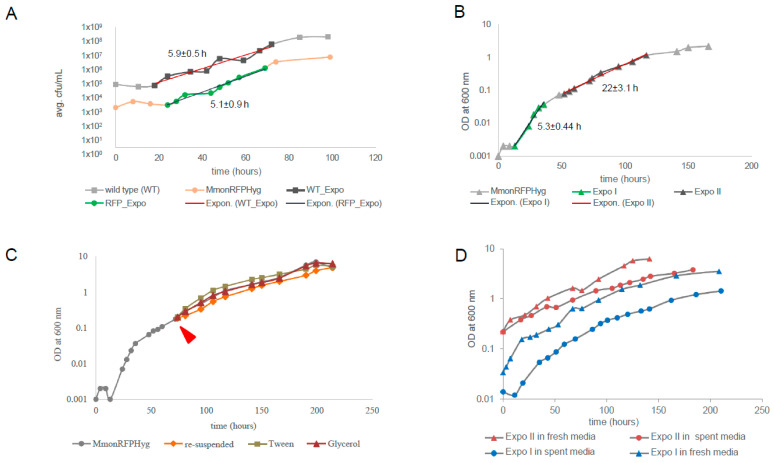
The representative growth curves for *Mmon*^T^ and *Mmon*^RFPHyg^ cultivated under various conditions. (**A**) The growth curve for *Mmon*^T^ (marked with squares) and *Mmon*^RFPHyg^ (marked with circles) cultivated on 7H10 (supplemented with hygromycin in the case of *Mmon*^RFPHyg^) plates with average generation times ± deviations in hours (h). WT_Expo and RFP_Expo represent exponential growth phases and the trend lines representing the slope of the growth curves are marked in red. The growth was generated by plotting the average values for each time point. The average CFU/mL was calculated from two biological replicates for each strain (see Section 2) and the generation times are given in hours ± error range. (**B**) The growth curve for *Mmon*^RFPHyg^ in liquid 7H9 media with two exponential growth phases, Expo I and Expo II (highlighted dark) and trend lines representing the slope of the curve in red used to calculate the generation times (see Section 2) ± error range as indicated. (**C**) The “normal” growth curve for *Mmon*^RFPHyg^ (in grey) and the growth curves after subjecting Expo II cells to various conditions, “re-suspended” in orange, addition of “Tween” in green, and addition of “glycerol” in red (for details see Section 2). The red arrow marks the time when cells were harvested and subjected to various conditions as outlined in see Section 2. (**C**,**D**) show representative best fit curves (with R^2^-values ≈ 1) to the observed bi-phasic growth pattern (see also Appendix A). (**D**) The growth curves for Expo I cells (*Mmon*^RFPHyg^) inoculated into “fresh media” (blue triangles) and “spent media” (blue circles) and for Expo II cells inoculated into “fresh media” (red triangles) and “spent media” (red circles).

**Figure 2 microorganisms-13-00475-f002:**
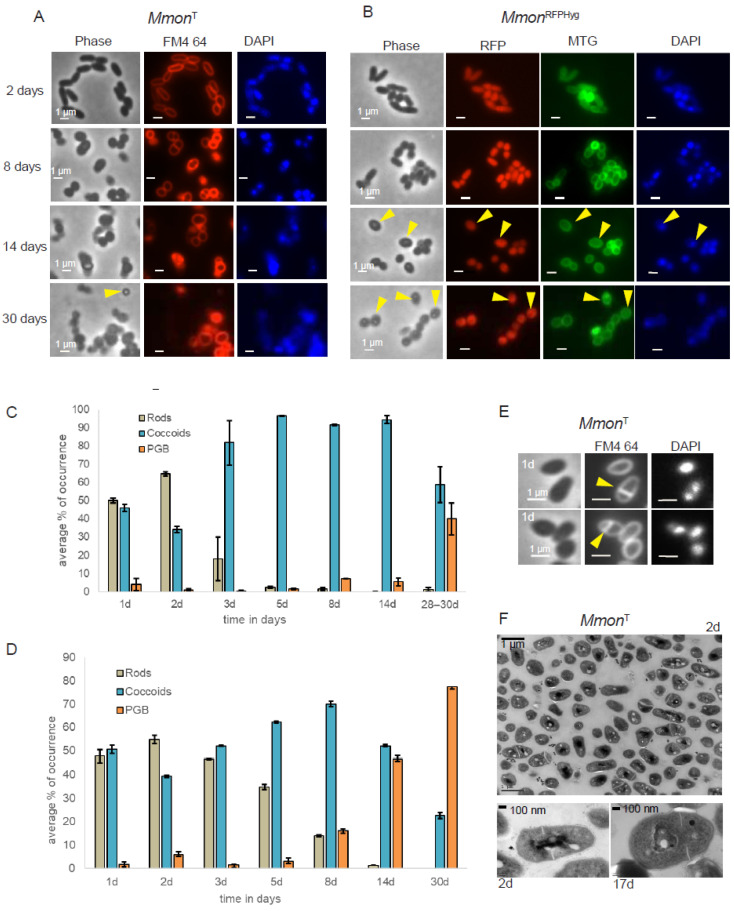
Microscopy of *Mmon*^T^ and *Mmon*^RFPHyg^ grown on 7H10 media and 37 °C. (**A**) Cells stained with FM4-64 (red) and DAPI (blue). Yellow arrows represent the PGB morphologies observed at different time points. Scale bar = 1 µm. (**B**) *Mmon*^RFPHyg^ cells expressing RFP (red) stained with MTG (green) and DAPI (blue). Yellow arrows indicate the PGB cell morphology. Scale bar = 1 µm. (**C**,**D**) Statistical representation of average percentage of occurrence of the various cell morphology types, *Mmon*^T^ (**C**) and *Mmon*^RFPHyg^ (**D**). A minimum of 350 cells were analysed for each time point. “d” = number of days. (**E**) One day old *Mmon*^T^ cells (rods and coccoids, yellow arrows mark the division site) stained with FM4-64 and DAPI depicting asymmetric septum formation (see **top** panel). Scale bar = 1 mm. (**F**) TEM image of a two days old culture showing dividing cells in the top panel. The bottom panel shows a TEM image of 2 and 17 days old coccoid *Mmon*^T^ cells showing asymmetric division sites. For each time point and condition, a minimum of 350 cells were counted. Error bars represent standard deviation. Plots represent the final averages of percentage of occurrence (see Section 2 for details).

**Figure 3 microorganisms-13-00475-f003:**
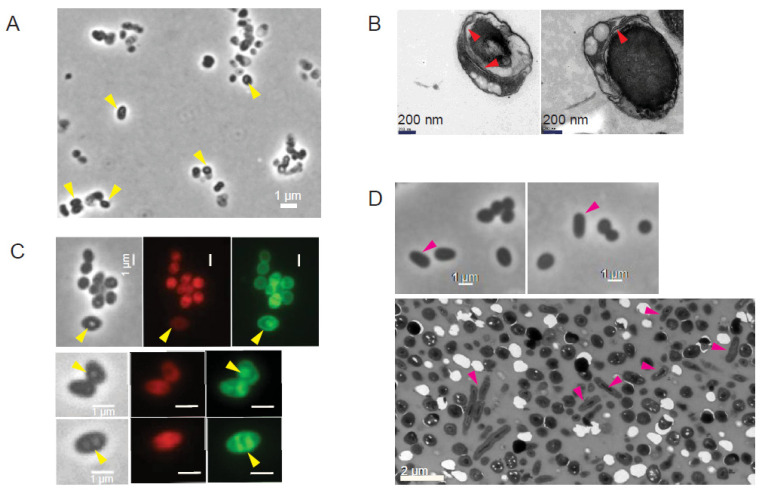
Visualisation of old *Mmon*^T^ and *Mmon*^RFPHyg^ cells grown on 7H10 media at 37 °C by phase contrast and transmission electron microscopy as indicated. (**A**) Phase contrast microscopy of 28 days old *Mmon*^T^ cells after enrichment for spores (see Section 2). Yellow arrows mark refractive PGB cells. Scale bar = 1 mm. (**B**) TEM images of 28 days old *Mmon*^T^ cells after spore enrichment. Red arrows mark internal membrane structures. (**C**) Internal structures (yellow arrows) observed in PGB cells detected in old *Mmon*^RFPHyg^ cultures (48 days old, **top** row; 14 days old, **middle** and **bottom** rows). Cells were stained with MTG (green) while red is the result of the presence of *rfp*. Scale bar = 1 mm. (**D**) *Mmon*^T^ cells one week after growth of enriched PGB cells on fresh 7H10 media. Phase contrast microscopy (**top** panels; Scale bar = 1 mm) and TEM (**bottom** panel; Scale bar = 2 mm). Pink arrows mark appearance of rod-shaped cells, see (**A**) for comparison.

**Figure 4 microorganisms-13-00475-f004:**
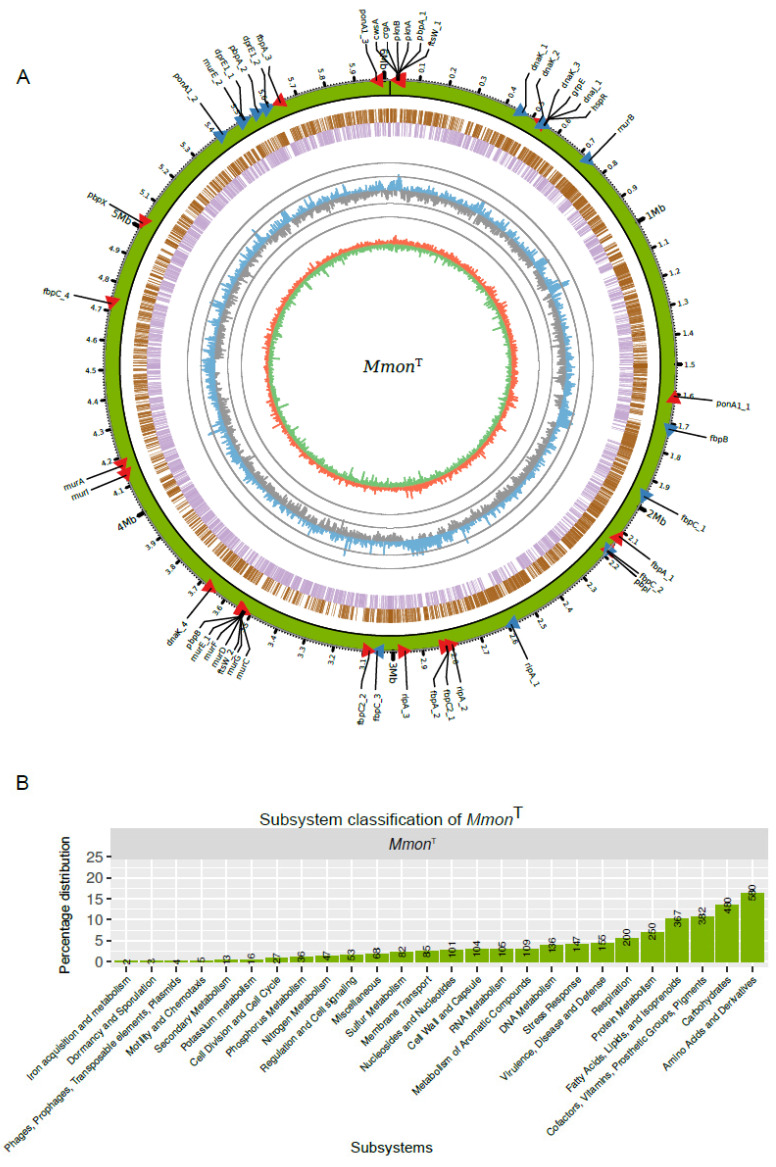
*Mmon*^T^ genome and functional classification of genes. (**A**) Overview of the *Mmon*^T^ complete genome. From the outer to inner circle: Green track illustrates the genome overlapping with scale along the genome length and position of genes (red and blue arrow heads mark the direction of transcription) listed in Appendix A. The next two circles represent genes in forward (brown) and reversed (purple) strands. The next circle shows the GC-content distribution calculated with a sliding window of 1000 bp, blue (higher than mean value) and grey (lower than mean value) “spikes” correspond to variations of the mean GC-content 68.4% in ±10 and ±20 units, i.e., outer grey circle = 88.4% and inner grey circle = 48.4%. The inner circle, red (positive) and green (negative) correspond to the GC-skew obtained using a sliding window of 1000 bp. Generation of circus plot, see http://circos.ca (last accessed on 9 July 2019). (**B**) Subsystem classification of 3557 *Mmon*^T^ genes as indicated.

**Figure 5 microorganisms-13-00475-f005:**
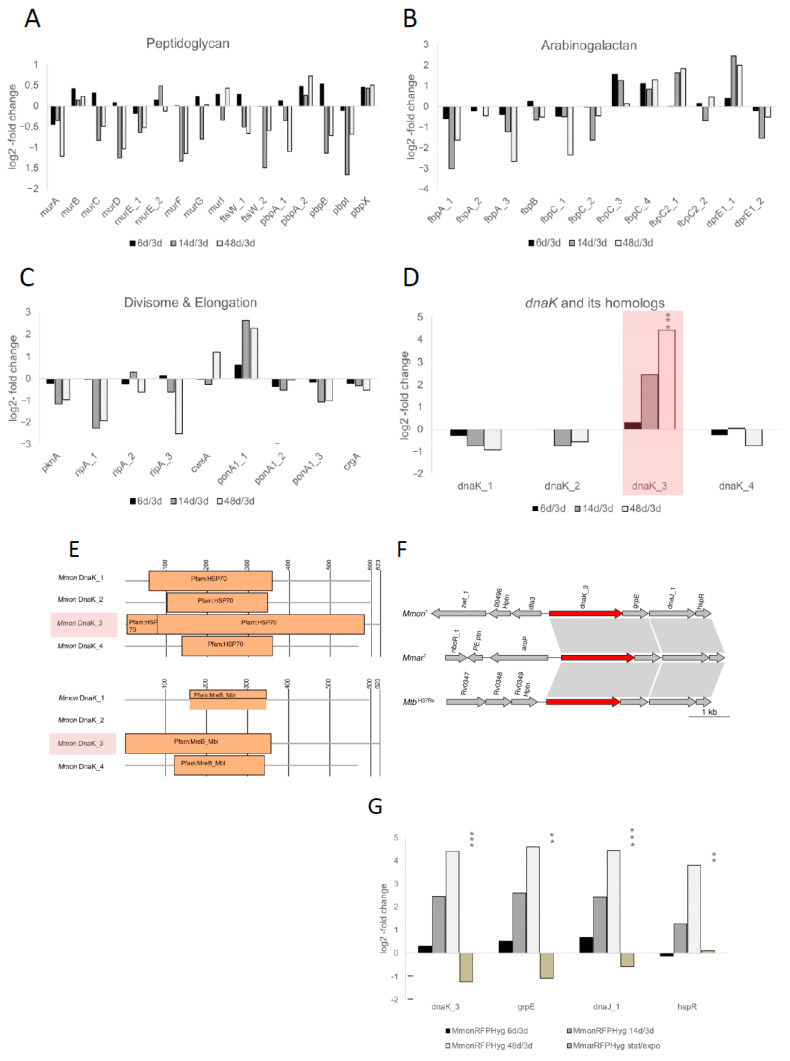
Analysis of mRNA levels of selected genes and *dnaK* paralogs in *Mmon*^T^. (**A**–**D**) Change in mRNA levels as indicated comparing *Mmon*^RFPHyg^ cells of different ages (6, 14 and 48 days) relative to 3 days old cells. The changes are expressed as log_2_-fold change and the selected genes were grouped as indicated. (**A**) Peptidoglycan-associated genes, (**B**) divisome- and elongation-associated genes, (**C**) arabinogalactan-associated genes and (**D**) *dnaK* homologs, the highlighted bars correspond to the change observed for *dnaK*_3. (**E**) Illustration of the domain architecture of the four DnaK proteins in *Mmon*^T^. Along with the expected Hsp70 domain, DnaK_3 (highlighted) is suggested to carry a MreB-Mbl domain while the other three lack this element. Notably, according to the Pfam database the average length of MreB-Mbl proteins encompass 313 amino acids. (**F**) Gene synteny for *dnaK*_3, *grpE*, *dnaJ* and *hspR* in *Mmon*^T^, *Mmar*^T^ and *Mtb*^H37Rv^. The arrows represent the genes as indicated. (**G**) Change in DnaK_3, GrpE, DnaJ and HspR mRNA levels expressed as log2-fold change as indicated. For *Mmon*^RFPHyg^, mRNA levels at different time points (6, 14 and 48 days of growth) relative to levels in exponentially growing cells (3 days of growth). In the case of *Mmar*^RFPHyg^ mRNA levels in exponentially growing cells (OD_600_ = 0.5) compared to levels in stationary cells (OD_600_ ≈ 3). Statistical significance, ** *p* < 0.01; *** *p* < 0.001.

**Figure 6 microorganisms-13-00475-f006:**
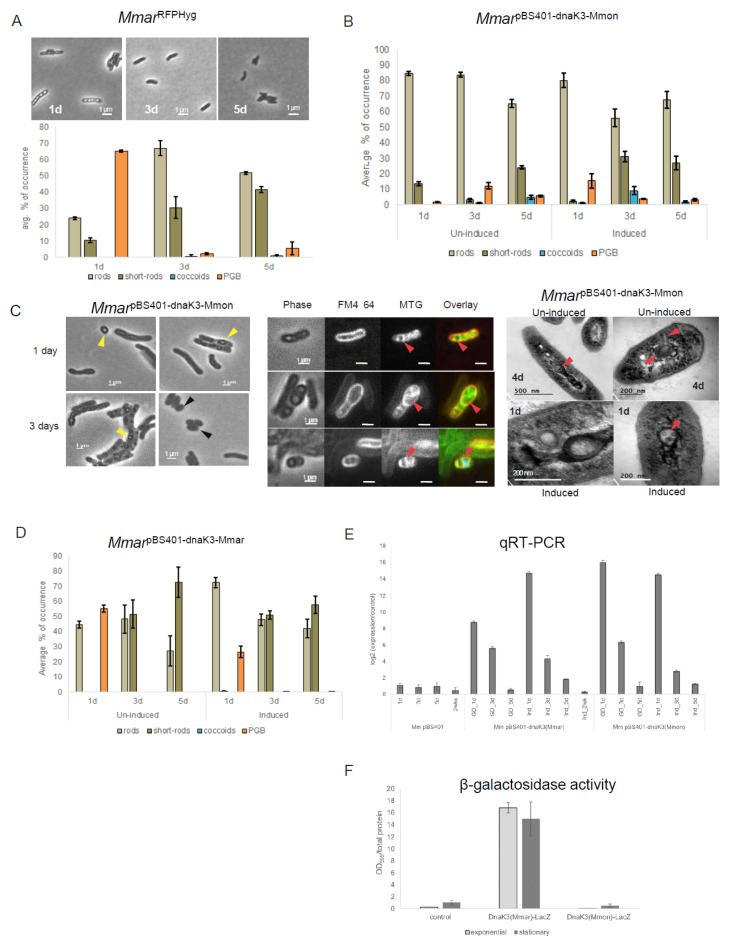
Analysis of the over-expression of *dnaK_3*^Mmon^ and *dnaK*_3^Mmar^ in *Mmar*^T^. (**A**) Represents time course microscopy of *Mmar*^RFPHyg^ (**top** panel) with the corresponding average frequencies of occurrence of the different cell morphotypes (**bottom** panel). Scale bar = 1 mm. (**B**) Statistical distribution of the different cell morphologies in *Mmar*^pBS401-dnaK3Mmon^ (un-induced and induced conditions) observed in the cell cultures. (**C**) Staining and microscopy of samples corresponding to (**B**). **Left** panel: microscopy of *Mmar*^pBS401-dnaK3Mmon^ at two time points (1 day and 3 days) exhibiting occurrence of PGB cells (yellow arrows) and coccoids (black arrows) under un-induced condition. Scale bar = 1 mm. **Middle** panel: MTG and FM4-64 staining of *Mmar*^pBS401-dnaK3Mmon^ cells showing internal membrane formation and compartmentalisation (red arrows). Scale bar = 1 mm. **Right** panel: TEM images of 4 days old *Mmar*^pBS401-dnaK3Mmon^ cells (un-induced and after 24 h induction). Red arrows mark the internal membrane formation and presence of internal structures. All cultures were grown on 7H10 media supplemented with hygromycin (100 μg mL^−1^) at 30 °C. (**D**) Statistical distribution of the different cell morphologies in *Mmar*^pBS401-dnaK3Mmar^ (un-induced and ‘Tet’ induced conditions) observed in the cell cultures. (**E**) Expression of *dnaK*_3^Mmon^ carried on pBS401 in *Mmar*^T^ as determined by qPCR. Cell extracts from three different time points (1 day, 3 days and 5 days) with and without tetracycline induction were considered, “d” = days. The log_2_-fold change was normalised to the “1 day-induction”. (**F**) Analysis of the expression of DnaK_3^Mmar^ and DnaK_3^Mmon^ in *Mmar*^T^ by β-galactosidase assay. β-galactosidase activity (DnaK_3-LacZ fused protein) upon addition of the CPRG substrate with (substrate) was measured using a spectrophotometer at 595 nm. The measured absorbance normalised with the total protein content for each sample as shown in the plots. The samples were protein extracts from *Mmar* cells carrying pIGN vector with *dnaK3*^Mmar^-*lacZ* or *dnaK3*^Mmon^-*lacZ*. The empty vector was used as control (see Section 2 for details). The numbers represent an average based on two independent experiments (round 1 and round 2) with two biological replicates.

**Table 1 microorganisms-13-00475-t001:** Compilation of the colony morphology characteristics of *Mmon*^T^ grown on different media. Observations were made after 15 days of growth at 37 °C.

Growth in Solid Media		
Strain	Media	Colony Morphology After 15 Days of Growth
*Mmon* ^T^	7H10 ***	pale yellow and smooth
	AK ***	pale yellow and smooth
	mG **	yellow and smooth
	LA **	yellow and smooth
	PDA *	smooth and yellow

*** good growth, ** moderate growth, * very poor or no growth.

**Table 2 microorganisms-13-00475-t002:** Compilation of OD_600_ measurements for *Mmon*^T^ derivatives in various liquid media supplemented with different antibiotics as indicated. The measurements were made after 5 days of growth at 37 °C.

Growth in Liquid Media		
Strain	Media	OD_600_ After 5 Days
*Mmon* ^T^	7H9	0.0–0.004
	LB ^#^	1.0–1.63
	mG ^#^	2.4–3.0
*Mmon* ^RFPHyg^	7H9 + hyg	0.72–1.5
*Mmon* ^RFPKan^	7H9 + kan	0.0
*Mmon* ^pBS401^	7H9 + hyg	3.5–4.0

^#^ growth with clumping making the OD_600_ measurements unreliable.

## Data Availability

The original contributions presented in this study are included in the article/Appendix A. Further inquiries can be directed to the corresponding author.

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
