# Peer review of "Age-Dependent Pleomorphism in Mycobacterium monacense Cultures"

_microorganisms, 2025, doi:10.3390/microorganisms13030475_

Round 1
Reviewer 1 Report
Comments and Suggestions for Authors
Introduction lacks pathogenic spectrum and prognosis of this human pathogen. There is an unnesessential focus on leprae and tuberculosis. There is no rational as to how the shape change affects disease transmission.
As authors have cloned a drug resistance gene in a human pathogen. This is a gain of function study needing IBC approval and containment. Authors need to update the manuscript with IBC approval and IBC facilities in the methods section. Transportation protocol for viable cultures between facilities needs to be disclosed.
CFU is to be written in all caps, as it is an abbreviation. Authors need to define all abbreviations at first instance.
Authors need to see if PGB states are more stress resistant. Not just genetic factors accumulation of metabolites could also be responsible for the shape change. Was aged media/cells analysed for metabolite constituents? Genetic expression change could be due to VBNC/dormant-like profile. Or these PGBs are more spore-like?
Authors have too much unreferenced speculative text. The paper is too long, and it's difficult to comprehend facts and differentiate from the author's openings. If not referenced, authors need to make this writeup concise for general understanding.
Author Response
Response to the reviewers comments and our response marked in blue.
See also the marked revised manuscript where the revised text is marked in red.
Reviewer1:
- Introduction lacks pathogenic spectrum and prognosis of this human pathogen.
Response - We emphasize that this is not a clinical report hence we have not included pathogenic spectrum and prognosis of M. monacense. And we state in the introduction "Hence, we decided to investigate growth, morphological characteristics and search for genes that might influence the cell shape of this RGM." Also, at the end of the discussion we have added "In conclusion, our findings provide insight into the biology of M. monacense and the possible role of the chaperone DnaK_3. As such, they may in identifying biomarkers that could be useful in the process to develop diagnostic tools and new drugs targeting mycobacteria."
- There is an unnesessential focus on leprae and tuberculosis.
Response – We argue that it is relevant to mention M. tuberculosis and M. leprae in a historical perspective in the introduction
- There is no rational as to how the shape change affects disease transmission.
Response – We have no information how the cell shape affects disease transmission.
- As authors have cloned a drug resistance gene in a human pathogen. This is a gain of function study needing IBC approval and containment. Authors need to update the manuscript with IBC approval and IBC facilities in the methods section. Transportation protocol for viable cultures between facilities needs to be disclosed.
Response – This has been clarified in the revised manuscript, see the beginning of the Materials and Methods section under 2.1.
- CFU is to be written in all caps, as it is an abbreviation. Authors need to define all abbreviations at first instance.
Response – This has been corrected, thank you.
- Authors need to see if PGB states are more stress resistant.
Response – This was checked see top of page 19 and the discussion, top page 31.
- Not just genetic factors accumulation of metabolites could also be responsible for the shape change. Was aged media/cells analysed for metabolite constituents?
Response – No we did not analyse this, but we discuss pppGpp in the discussion, top page 29.
- Genetic expression change could be due to VBNC/dormant-like profile. Or these PGBs are more spore-like?
Response – The PGBs appear as spore-like but as stated on top of page 19 they did not meet the requirements of endospores. We also state "These data suggested that the Mmon spore-like PGB structures are different compared to Mmar and MAP spores.".
- Authors have too much unreferenced speculative text.
Response – Very little information is known about M. monacense and we argue that we refer to the best of our knowledge to publications relevant to our discussion.
- The paper is too long, and it's difficult to comprehend facts and differentiate from the author's openings. If not referenced, authors need to make this writeup concise for general understanding.
Response – See above and we revised the manuscript to be more succinct and we added headlines for the sections in the discussion to make it easier to read.
----------------------------------------------------------------------
Reviewer2:
- The clear identification of: a) aim of the study and b) the scientific hypothesis of this study are required for understanding of the significance of this work
Response – We have revised the introduction to clarify the aim of the study, see introduction.
- The clear explanation of selected methods for Mycobacteria study is required for study design understanding. It would be a large benefit, if you will explain the practical outcomes from your work.
Response – We use selected methods and standard microscopy to study mycobacteria as described in the Materials and Methods. See also e.g. Ghosh et al. 2009 and Singh et al. 2013 (reference number 14 and 25, respectively).
- Table 1A - more clear explanation of the colonies' morphology would be
Beneficial
Response – The given description is what we observe.
- Fig. 2 - the clear identification of control parameters/baseline for
study would be sufficient for comparison - i.e. reference method or some
standard.
Response – The figure describes what we observe and since M. monacense have not, to the best of our knowledge, previously been studied in this way we do not have access to any reference or standard.
- Fig. 3 - the signs has to be more detailed.
Response – The signs are clear in our opinion.
- Fig. 5 is very complicated and has to be subdivided to several figures
and they have to be explained deeper.
Response – We argue that these panels should be together since they are part of our analysis of the gene identification and RNASeq data.
- Fig. 6 - same comment.
Response – See above, similar response.
- Separate conclusion section would be a benefit.
Response – We have added headlines 4.1 to 4.4 in the discussion and 4.4 refers to "Concluding remarks".
- The references list require to be added by modern sources (2020-24).
Response – The reference list includes to the best of our knowledge the appropriate references.
--------------------------------------------------------------------
Currently, we have set the deadline to 7 February 2025. Looking forward
to receiving your revised manuscript.

Reviewer 2 Report
Comments and Suggestions for Authors
- The clear identification of: a) aim of the study and b) the scientific hypothesis of this study are required for understanding of the significance of this work
- The clear explanation of selected methods for Mycobacteria study is required for study design understanding. It would be a large benefit, if you will explain the practical outcomes from your work.
- Table 1A - more clear explanation of the colonies' morphology would be beneficial
- Fig. 2 - the clear identification of control parameters/baseline for study would be sufficient for comparison - i.e. reference method or some standard.
- Fig. 3 - the signs has to be more detailed.
- Fig. 5 is very complicated and has to be subdivided to several figures and they have to be explained deeper.
- Fig. 6 - same comment.
- Separate conclusion section would be a benefit.
- The references list require to be added by modern sources (2020-24).
Comments on the Quality of English LanguageEnglish could be improved.
Author Response

(The authors gave the same response as above.)

Round 2
Reviewer 1 Report
Comments and Suggestions for Authors
Authors have addressed all the comments raised by me. The revised manuscript is fit for publication in MDPI